# Updates and Future Directions for the Nottingham Research Programme on Primary Breast Cancer in Older Women

**DOI:** 10.3390/cancers17030346

**Published:** 2025-01-21

**Authors:** Ruth Mary Parks, Kwok-Leung Cheung

**Affiliations:** Nottingham Breast Cancer Research Centre, University of Nottingham, Nottingham NG7 2RD, UK; kl.cheung@nottingham.ac.uk

**Keywords:** breast cancer, surgery, geriatric oncology, older women

## Abstract

The Nottingham programme on primary breast cancer in older women is a unique resource, appearing to be the only one of its kind reported in the literature. In this update, we will describe the background to this subject, the main findings from the research team to date, focusing on biology, psycho-social aspects, and cost-effectiveness of different treatment modalities in this cohort, and directions for future research in this field.

## 1. Introduction

The risk of breast cancer increases with age. The general population is ageing [1], and, therefore, it is estimated that, by 2040, the number of patients diagnosed with breast cancer per year will double. Around 40% of these diagnoses will be older women (≥70 years) [2]. Despite this stark reality, there are few published treatment guidelines specific to older women with breast cancer that recognise their unique biological differences and healthcare needs.

Updated recommendations on the management of older women with breast cancer have been developed by a taskforce of European Society of Breast Cancer (EUSOMA) and International Society of Geriatric Oncology (SIOG) specialists, which were published in 2021 [3]. These guidelines highlight that decision making in older women should not be guided by age but should involve careful consideration of life expectancy, taking into account competing risks of mortality and geriatric assessment where appropriate. Patient preferences should be assessed at each stage of the treatment pathway, including screening, management of ductal carcinoma in situ, decision for primary endocrine therapy, surgery, radiotherapy, adjuvant systemic therapy, and in the secondary settings. Major updates to the recommendations since a previous publication [4] include guidance on screening for frailty and geriatric assessment, the inclusion of toxicity calculators when considering systemic therapies, the consideration of the cultural and social needs of the patient, and the use of multigene genomic assays and neoadjuvant systemic therapy.

There are two important UK studies specific to primary breast cancer in older women that provide a snapshot as to what is happening currently in clinical practice in this cohort, alongside guidance for managing these patients.

Bridging the Age Gap (BTAG) was a prospective, multicentre, observational cohort study collecting data from 56 participating centres in the UK between 2013 and 2018. The aim of the study was to capture real-world data from over 3000 women aged > 70 years of age with early operable primary breast cancer [5,6]. The study captured personal and cancer characteristics of these women as well as the treatments received and long-term outcomes. The data were used to design a risk calculator and decision support aids for patients who have a choice between surgery and endocrine therapy in the primary setting and the decision of chemotherapy in the adjuvant setting [7]. The Age Gap decision tool can be used by clinicians in the hospital setting to provide additional information to patients on their predicted outcomes with different treatment types. This can be used in conjunction with the multi-disciplinary team and other standard support methods available.

The National Audit of Breast Cancer in Older Patients (NABCOP) evaluates processes, care, and outcomes of older women treated for breast cancer in the UK and produces annual reports, the latest of which was published in 2022 [8]. This audit provides “real-world” data on what is happening in clinical practice and how this might differ from the guidelines, for example, by including the number of patients undergoing non-surgical treatments.

Some challenges remain in managing older women with breast cancer. Two specific areas of interest to the Nottingham research group are (1) when to offer endocrine therapy (PET) as an alternative option to surgery in the primary setting and (2) when or when not to give adjuvant therapy.

The adjuvant landscape has changed significantly in the past few years, and there are now many more potential adjuvant (and, thereby, neoadjuvant) therapies which may be offered to older women, including chemotherapy, radiotherapy, endocrine therapy, bisphosphonates, CDK 4/6 inhibitors, immunotherapy agents, and newer anti-HER2 therapies. Each additional treatment has risks and benefits that must be discussed with the patient, specifically the potential for increased toxicity with, sometimes, minimum benefits in terms of survival outcomes [9].

## 2. Contrast Between Breast Cancer in Older Compared to Younger Women, Supported by Our Research Publications, and Implications for Future Works

Briefly, the research programme has three main aims, as shown in Figure 1: (1) to explore the unique biological differences in breast cancer in older compared to younger women, (2) to explore unique psycho-social factors important to this patient population through geriatric assessment and other measures, such as quality of life (QOL) assessment, which may impact treatment decisions, and (3) to compare the cost-effectiveness between different therapeutic strategies.

### 2.1. Biology

The Nottingham database contains 1757 consecutive patients who attended the Nottingham Breast Institute from 1973 to 2010, with clinical follow-up information available until November 2024. These patients were diagnosed with early-stage primary breast cancer which was operable at diagnosis [10,11]. There is a comparative younger cohort of primary breast cancer patients also from Nottingham, available for comparison [12].

From the whole cohort of 1757 patients, 813 underwent surgery as the primary treatment. From the surgical samples, tissue microarrays (TMAs) were constructed from 575 specimens, and a panel of 25 biomarkers was measured [10]. It was also possible to construct TMAs from core needle biopsy (CNB) samples that had been taken from 693 patients at diagnosis. An array of 17 biomarkers was analysed in these TMAs in ER-positive cases [13].

A cluster analysis was performed in both the surgical TMAs and the CNB TMAs. This identified a unique cluster of disease, distinct from the known standard clusters of breast cancer seen in younger women [10,13]. The unique cluster was named “low ER luminal” and had high expression of luminal cytokeratin (CK)s, mucin (MUC)1, and the human epidermal growth factor receptor (HER)3. The low ER luminal cluster also had a distinct breast cancer-specific survival pattern.

Further works looking at biology according to age also found differences in the subtypes of HER2-positive [14] and triple-negative breast cancer [15] and in the treatment response [16].

The work described above identified potential biomarkers that may be of use in predicting outcomes depending on therapy (surgery, PET, adjuvant endocrine therapy) and survival in older women with primary breast cancer, including novel markers not currently used in clinical practice such as cytoplastic cyclin-E [17] and liver kinase-B1 [18].

### 2.2. Psycho-Social Aspects

The second arm of the research programme includes a prospective study implementing comprehensive geriatric assessment (CGA), alongside QOL assessments, in older women with primary breast cancer to determine whether parts of CGA can be implemented to help decision making.

The pilot study results were published [19], finding that an increasing age (≥80 years), greater number and impact of comorbidity (≥4), an increasing number of medications taken on a daily basis (≥4), and a slower timed up-and-go test (≥19 s) were associated with a greater likelihood of the patient undergoing non-surgical treatment rather than surgery. Significant associations between QOL measures and the decision to have non-surgical treatment were also found [20], while another study expanded the research area to include international centres [21].

Another area of interest to the team is looking at the disparity between the surgical treatments offered to older women: for example, the provision of reconstruction. Our group has shown through systematic reviews that there is a disparity between the uptake of postmastectomy immediate breast reconstruction [22] and oncoplastic breast conserving surgery in older versus younger women [23]. This is replicated in real-world data that have been obtained from working with EUSOMA [24]. Through our systematic reviews, there appears to be a gap in knowledge as to the exact reasons for this disparity, but they appear to be multi-factorial and include patient factors such as comorbidity and wishes, physician factors such as pre-existing bias, and system factors such as the logistics involved in multiple hospital visits.

### 2.3. Cost-Effectiveness

The third arm of the research programme is investigating the cost-effectiveness of differing treatment modalities.

A systematic review looking at model-based economic evaluation of treatments for older women with primary breast cancer was conducted to investigate the following parameters: health-related QOL, treatment effect, natural history, and resource use [25]. The review concluded that a better understanding of the value patients place on different treatments was required to help improve health outcomes, as well as aid decision making and resource allocation by healthcare providers.

A second review with a meta-regression analysis aimed to identify the use of health state utility values as measured by validated health-related QOL measures in older women with early breast cancer [26]. Overall, the findings support surgery to be more cost-effective than PET, but this difference diminishes with increasing frailty of the patient.

These findings are important for healthcare providers to consider, especially in view of an ageing population.

## 3. Important Future Areas of Research and Contributions of Our Research Team

Further work is underway to examine a complete panel of around 30 biomarkers in all SE TMAs and CNB TMAs available from the Nottingham cohort. The aim of this work would be to develop a prognostic algorithm that could be used to help determine outcomes if a patient was considering different therapeutic strategies. This concept could also be applied to those considering adjuvant therapies.

Recruitment to the prospective study on CGA is ongoing, with a target of 1558 patients, to determine any statistically significant findings. The aim of this would be to generate a CGA specifically for older women with breast cancer to aid treatment decision making.

Further work looking into the reasons for the inequality in the provision of reconstruction between younger and older women is underway.

There is no doubt that the number of older women living with breast cancer will increase; the global population is ageing, and, with an increasing life expectancy, we can expect more breast cancer cases to be diagnosed, particularly in older women, as age is the biggest risk factor for breast cancer after a female gender. Every attempt should be made to provide personalised care to these patients to maintain quality of life whilst providing optimal cancer care. The involvement of older women in the process of shared decision making is a necessity. In addition to the unique challenges presented by the disease itself, this cohort has different social and cultural expectations.

## Figures and Tables

**Figure 1 cancers-17-00346-f001:**
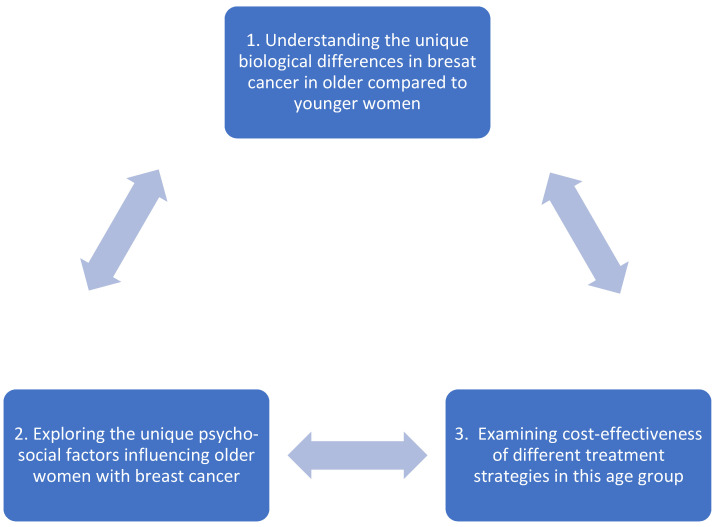
An overview of the aims of the Nottingham research team on primary breast cancer in older women.

## Data Availability

No new data have been presented in this paper.

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
