# Peer review of "Updates and Future Directions for the Nottingham Research Programme on Primary Breast Cancer in Older Women"

_cancers, 2025, doi:10.3390/cancers17030346_

Round 1
Reviewer 1 Report
Comments and Suggestions for Authors
As the proportion of elderly individuals in developed countries continues to rise, the significance of treating breast cancer in this demographic is expected to grow accordingly. This paper elucidates the background, findings, and future research directions of The Primary Breast Cancer in Older Women Programme in Nottingham. The programme stands out as highly intriguing, demonstrating consistent and significant results. It is crucial for medical professionals involved in breast cancer care to be well-informed about this programme.
Author Response
Thank you for your encouraging comments on this paper.
Reviewer 2 Report
Comments and Suggestions for Authors
The issue of breast cancer in older women is becoming increasingly important in aging societies. The authors, who are experts in this field and have an extensive publication list, particularly in the area of breast cancer in older women, present the Nottingham research program on primary breast cancer in older women in their well-written article. They briefly summarize the background and rationale for this research program and cover the biology, psychosocial aspects, and cost-effectiveness of older breast cancer patients. They then conclude their short manuscript with future research areas. These findings are interesting and the scientific community is very interested in further important results in this important patient cohort.
Author Response

(The authors gave the same response as above.)

Reviewer 3 Report
Comments and Suggestions for Authors
The work is of international interest and is excellently written.
Two considerations
a) I suggest to better explain in the conclusion: the increase in average life will lead to an increase in breast cancer in women, especially in old age;
b) the involvement of women in the therapeutic choice will be a necessity for older women because in addition to the severity of the disease, the social and cultural context and its expectations will have to be taken into account.
2 clarifications:
a) line 79: 1700 consecutive patients; line 84: 1757 patients: please explain the number better because it is confusing.
b) line 120: I do not agree that the reasons for this disparity are not known. Older women suffer a prejudice related to the fact that because of their age they are more often thought of postponing or avoiding reconstruction. At least give a possible hypothesis.
Author Response
Thank you for your comments which we have addressed as follows:
a) I suggest to better explain in the conclusion: the increase in average life will lead to an increase in breast cancer in women, especially in old age; The conclusion has been amended to reflect this
b) the involvement of women in the therapeutic choice will be a necessity for older women because in addition to the severity of the disease, the social and cultural context and its expectations will have to be taken into account. This has been added to the concluding statement.
2 clarifications:
a) line 79: 1700 consecutive patients; line 84: 1757 patients: please explain the number better because it is confusing. This has been amended
b) line 120: I do not agree that the reasons for this disparity are not known. Older women suffer a prejudice related to the fact that because of their age they are more often thought of postponing or avoiding reconstruction. At least give a possible hypothesis. This section has been amended and expanded
Reviewer 4 Report
Comments and Suggestions for Authors
In this commentary, the authors present a short update and review on the aims and results of the studies on breast cancer in older women in Nottingham. As the Nottingham centre is an important institution and this kind of research, integrating biomarkers, social and economical aspects it is a good idea to distribute these ideas further into the community.
The manuscript is also well written.
Nevertheless, if possible, a figure better illustrating the research concept further would be a plus. This could also be done as graphical abstract.
Author Response
Thank you for your comments. A figure has been added.